# Genome Mining Reveals a Surprising Number of Sugar Reductases in *Aspergillus niger*

**DOI:** 10.3390/jof9121138

**Published:** 2023-11-24

**Authors:** Astrid Mueller, Li Xu, Claudia Heine, Tila Flach, Miia R. Mäkelä, Ronald P. de Vries

**Affiliations:** 1Fungal Physiology, Westerdijk Fungal Biodiversity Institute & Fungal Molecular Physiology, Utrecht University, Uppsalalaan 8, 3584 CT Utrecht, The Netherlands; a.mueller@wi.knaw.nl (A.M.); l.xu@wi.knaw.nl (L.X.); claudia.heine@hotmail.com (C.H.); tdflach@outlook.com (T.F.); 2Department of Microbiology, University of Helsinki, Viikinkaari 9, 00014 Helsinki, Finland; miia.r.makela@helsinki.fi

**Keywords:** sugar catabolism, orthology-based approach, metabolic engineering, *Aspergillus niger*, enzyme characterization, synthetic biology

## Abstract

Metabolic engineering of filamentous fungi has received increasing attention in recent years, especially in the context of creating better industrial fungal cell factories to produce a wide range of valuable enzymes and metabolites from plant biomass. Recent studies into the pentose catabolic pathway (PCP) in *Aspergillus niger* have revealed functional redundancy in most of the pathway steps. In this study, a closer examination of the *A. niger* genome revealed five additional paralogs for the three original pentose reductases (LarA, XyrA, XyrB). Analysis of these genes using phylogeny, in vitro and in vivo functional analysis of the enzymes, and gene expression revealed that all can functionally replace LarA, XyrA, and XyrB. However, they are also active on several other sugars, suggesting a role for them in other pathways. This study therefore reveals the diversity of primary carbon metabolism in fungi, suggesting an intricate evolutionary process that distinguishes different species. In addition, through this study, the metabolic toolkit for synthetic biology and metabolic engineering of *A. niger* and other fungal cell factories has been expanded.

## 1. Introduction

In recent years, our understanding of fungal metabolism has strongly improved, revealing redundancy and diversity [1] in specific pathways. This has been especially well described for the pentose catabolic pathway (PCP) in *Aspergillus niger* [2]. The PCP is a crucial metabolic pathway in filamentous fungi for converting the monosaccharides L-arabinose and D-xylose into value-added compounds [3]. This pathway involves a series of oxidation, reduction, and phosphorylation reactions, ultimately leading to the formation of D-xylulose-5-phosphate, which enters the pentose phosphate pathway (PPP). In contrast to an initial simple description of the pathway, where each step is catalyzed by a single enzyme, recent studies have revealed a higher level of complexity and redundancy in its functioning, demonstrating the involvement of additional enzymes. Moreover, variations in fungi regarding the genes responsible for the conversion of D-xylose and L-arabinose have been demonstrated. For instance, *Trichoderma reesei* possesses only one D-xylose-reductase-encoding gene (*xyl1*) that catalyzes both initial steps [4].

In fungal cell factories, the PCP and its enzymes facilitate the production of chemicals and biofuels from lignocellulosic materials [5,6]. Using genetic engineering, abundant lignocellulose building blocks, like L-arabinose and D-xylose, can be transformed by microorganisms into, e.g., bioethanol [7], xylitol [8,9], and succinic acid [10].

To improve the efficiency of lignocellulose use, a comprehensive understanding of the PCP and its enzymes is essential. A previous study demonstrated that the first step in pentose catabolism, the conversion of D-xylose and L-arabinose into xylitol and L-arabitol, respectively, involves at least three pentose reductase enzymes in *A. niger*, the NADPH-dependent L-arabinose reductase (LarA, NRRL3_10050; EC 1.1.1.21) [11] and two D-xylose reductases (XyrA, NRRL3_1952; XyrB, NRRL3_10868; EC 1.1.1.307) [12,13]. The triple deletion mutant of these genes [14] did not significantly reduce the growth of *A. niger* on wheat bran (WB) and sugar beet pulp (SBP), which contain considerable amounts of L-arabinose and D-xylose. This may be due to the presence of other components in these substrates but could also indicate the involvement of additional genes that are expressed under these conditions but not during growth on L-arabinose or D-xylose.

In this study, we therefore delved deeper into this metabolic versatility by evaluating the presence of additional paralogs of the three known pentose reductases in the *A. niger* genome. The identified five candidates showed no-to-low expression on L-arabinose and D-xylose and were compared to LarA, XyrA, and XyrB based on biochemical properties and gene expression. To demonstrate their in vivo functionality, the ability of these candidates to compensate for the combined deletion of *larA*, *xyrA*, and *xyrB* was evaluated by inserting them behind the promoter of the deleted genes.

## 2. Materials and Methods

### 2.1. Bioinformatics and Phylogenetic Analysis

All amino acid sequences in this study were obtained from JGI MycoCosm (https://genome.jgi.doe.gov/mycocosm/home, accessed on 14 March 2023). The amino acid sequences of *A. niger* LarA NRRL3_10050 [11,13], XyrA NRRL3_1952, and XyrB NRRL3_10868 were used as a query for BlastP analysis to identify their homologs in various fungal species, including Eurotiomycetes (*A. niger*, *Aspergillus nidulans*, and *Penicillium subrubescens*), Sordariomycetes (*T. reesei*, *Neurospora crassa*, and *Podospora anserina*), Saccharomycotina (*Saccharomyces cerevisiae* and *Pichia pastoris*), Dothideomycetes (*Mycosphaerella graminicola* and *Stagonospora nodorum*), and Leotiomycetes (*Botrytis cinerea*). Sequences were aligned using MAFFT (http://ww.ebi.uk/Tools/msa/mafft/, accessed on 14 March 2023). Phylogenetic analysis was performed using maximum likelihood (ML), neighbor-joining (NJ), and minimal evolution (ME) methods implemented in Molecular Evolutionary Genetic Analysis software version 7 (MEGA7) with 95% partial deletion and the Poisson correction distance of substitution rates [15]. Statistical support for phylogenetic grouping was estimated using 500 bootstrap re-samplings. For the circular phylogenetic tree that included transcriptome data, protein sequences of selected species for which transcriptome data were available were aligned with MAFFT. The positions containing >20% gaps were removed from the alignment using trimAl V1.3 [16]. Next, phylogenetic analysis was performed with the maximum likelihood method using IQ-TREE 2 [17] with 1000 UFBoot2 bootstrapping and the MFP model. iTOL V6.8.1 [18] was used to visualize the tree. The transcriptome data were obtained from previous liquid culture studies in our group [19] for *A. niger*, *A. nidulans*, *P. subrubescens*, and *T. reesei*, while the *N. crassa* data were from Wu et al. [20].

### 2.2. Strain, Culture Conditions, and Media

*Escherichia coli* DH5α was used for all plasmid propagation and was grown in Luria–Bertani (LB) medium supplemented with an appropriate antibiotic. All strains used in this study were derived from *A. niger* CBS 138,852 (*cspA1*, *kusA*::*amdS*, *pyrA6*; [21]), which was obtained from the Westerdijk Fungal Biodiversity Institute culture collection (Utrecht, the Netherlands). In addition, this strain was used to construct the single-pathway deletion mutants (△*larA*, △*xyrA*, and △*xyrB*). The strains were maintained on *Aspergillus* Minimal Medium (MM) or Complete Medium (CM) [22] plates (1.5% *w*/*v* agar) at 30 °C with 1% D-glucose and 1.22 g/L uridine (Sigma Aldrich, Zwijndrecht, The Netherlands). An overview of the *A. niger* strains used in this study is provided in Table 1.

Solid-state cultivations for growth profiling were performed using MM plates containing 1.5% agar (*w*/*v*) with the addition of 25 mM monomeric substrates or 1% biomass substrates (*w*/*v*). All media were supplemented with 1.22 g/L of uridine. Growth-profiling plates were inoculated with 1000 freshly harvested spores in 2 µL of ACES buffer and incubated at 30 °C for up to 5 days. Growth was evaluated using visual inspection. For transfer experiments, pre-cultures containing 250 mL of CM with 2% D-fructose and 1.22 g/L of uridine in 1 L Erlenmeyer flasks were inoculated with 10^6^ spores/mL and incubated for 16 h. Mycelia were harvested using filtration, washed with MM, and transferred to 250 mL Erlenmeyer flasks containing 50 mL of MM supplemented with the appropriate carbon source and 1.22 g/L of uridine. All cultures were performed in triplicate. After incubation, the mycelia were harvested using vacuum filtration, dried between tissue paper, and frozen in liquid nitrogen. All samples were stored at −80 °C until being processed.

### 2.3. Molecular Cloning and Transformation in Escherichia coli Protein Production Strains

RNA was extracted from *A. niger* grown on sugar beet pulp using TRIzol reagent (Invitrogen, Uden, The Netherlans) and purified with NucleoSpin RNA (Macherey-Nagel, Landsmeer, The Netherlands). ThermoScript Reverse Transcriptase (Invitrogen) was used to obtain a total cDNA sample. Selected *A. niger* NRRL3 genes were amplified using specific primers from the generated cDNA (Appendix A). Purified PCR products were digested with the appropriate restriction enzymes (Invitrogen). After digestion, the products were ligated into pET-28a(+) and cloned using the Seamless Ligation Cloning Extract (SLiCE) cloning method [23]. The recombinant plasmid XyrB NRRL3_10868 was obtained from a previous study [13]. The resulting constructs were transformed into *E. coli* DH5α for propagation and sequencing. The plasmids were isolated and transformed into *E. coli* strains BL21 Star (DE3) *pLysS* (Invitrogen) and ArcticExpress (DE3; Agilent) according to the manufacturer’s instructions. Transformants were selected on LB medium containing 25 mg L^−1^ of kanamycin and 25 mg L^−1^ of gentamycin for BL21 Star (DE3) *pLYsS* and ArcticExpress (DE3), respectively. By using gene-specific primers, positive colonies were verified using colony PCR.

### 2.4. Production and Purification of Recombinant Proteins

Transformed *E. coli* strains were grown to an OD_600_ of 0.5 in LB medium containing the appropriate antibiotics at 37 °C and 250 rpm. IPTG at a final concentration of 0.1 mM was added, and the cultures were incubated overnight to induce protein production at 12 °C and 250 rpm. Cells were harvested using centrifugation at 11,000 rpm and 4 °C for 15 min, and the resulting pellets were resuspended in 30 mL of BugBuster Protein Extraction Reagent containing Benzonase Nuclease (Merc Millipore, Darmstadt, Germany). After incubation at 4 °C for 30 min, centrifugation at 11,000× *g* and 4 °C for 20 min was performed to remove cell debris. The supernatant was filtered through 45 µm syringe filters (Whatman, GE Healthcare Life Sciences, Pittsburgh, PA, USA) and applied to a 1 mL HisTrap FF column (GE Healthcare Life Sciences, Pittsburgh, PA, USA) at a flow rate of 1.0 mL min^−1^. The bound proteins were eluted with 20 mM HEPES, 400 mM imidazole, and 400 mM NaCl (pH 7.5) after washing with 20 mM HEPES, 20 mM imidazole, and 400 mM NaCl (pH 7.5). Fractions corresponding to the absorbance peak at 280 nm were collected and verified using SDS-PAGE and then pooled and desalted with 20 mM HEPES (pH 7.0) using an Amicon Ultra 15 mL Filter (Merck Millipore, Darmstadt, Germany) with an MW cutoff of 10 kDa. All purification steps were performed at 4 °C. Protein concentrations were determined using the Pierce BCA Protein Assay Kit (Thermo Fisher Scientific, Waltham, MA, USA).

### 2.5. Enzyme Assays

Sugar reductase activity was determined in Na-phosphate buffer at pH 7.0, with 0.2 mM NADPH and 100 mM substrate. For the kinetic analysis, D-xylose and L-arabinose reactions were performed in a reaction mixture of 0.2 mL total volume. The reaction mixtures contained 0.2 mM NADPH, different concentrations of D-xylose or L-arabinose (0–300 mM), and the appropriate amount of purified protein in HEPES buffer, pH 7.0. The conversion of NADPH (extinction coefficient ℇ = 6.22 × 10^−3^ M^−1^ cm^−1^) was monitored from the reaction mixture by measuring the decrease in absorbance at 340 nm in flat-bottom microtiter plates (Grainer Bio-One, Kremsmünster, Austria) in a microplate reader (FLUOstar OPTIMA, BMG LABTECH, Ortenberg, Germany). The kinetic constants K_m_ and V_max_ were calculated from the Michaelis–Menten equation fitted to the measured data.

### 2.6. Construction, Protoplast-Mediated Transformation, and Purification of Mutant Strains

CRISPR/Cas9 genome editing was performed using the ANEp8-Cas9-*pyrG* plasmid, which contains the *pyrG* gene as a selection marker [24]. Appendix A lists the guide sequences and primers used in this study. Phusion^TM^ High-Fidelity DNA Polymerase (Thermo Fisher Scientific) was used for PCR amplification, following the manufacturer’s instructions, and genomic DNA from the reference strain was used as a template. An overlapping specific gene sequence was incorporated into the downstream forward primer and the downstream reverse primer to match the to-be-swapped gene sequence of NRRL3_2854, NRRL3_7282, NRRL3_8020, NRRL3_10849, and NRRL3_6930. PCR was used to produce linear deletion DNA cassettes by fusing these two fragments. The PCR DNA and Gel Band Purification Kit (Promega, Leiden, The Netherlands) was used to purify the amplified swap cassettes. *A. niger* protoplasting was performed, as previously described [14]. Polyethylene glycol (PEG)-mediated transformation of *A. niger* protoplasts was performed, as described in detail in [25]. Assembled ANEp8-Cas9-gRNA plasmid DNA was used in combination with the purified linear deletion DNA cassette in the transformations. Five colonies per mutant were selected and streak-purified twice on MM plates supplemented with 25 mM D-glucose. For mutant confirmation, genomic DNA was isolated from the mycelia of putative gene swaps using the Wizard^®^ Genomic DNA Purification Kit (Promega). The flanking region and the area inside the swapped gene were amplified using diagnostic PCR to identify the correct mutants. Mutants were re-inoculated on MM plates with the appropriate carbon source and uridine and then on 5-FOA plates to remove the self-replicating plasmid.

### 2.7. Determination of Biomass

After transfer, mutants were grown for 3 days in liquid cultures containing 100 mM D-glucose, 100 mM D-xylose, 100 mM L-arabinose, or 50 mM D-xylose + 50 mM L-arabinose incubated in an orbital shaker at 250 rpm and 30 °C. Mycelia were harvested using vacuum filtration, dried between tissue paper, frozen in liquid nitrogen, and freeze-dried. Freeze-dried samples were weighed, and the three biological replicates were averaged for biomass calculations.

### 2.8. Preparation of Cell-Free Extracts and Enzyme Activity Assays

Frozen mycelial samples were disrupted with steel grinding balls for 1 min at frequency 25 s^−1^ with TissueLyser II (Qiagen, Venlo, The Netherlands) and resuspended in 1 mL of cold extraction buffer (50 mM K-phosphate at pH 7.0, 5 mM MgCl_2_, 5 mM 2-mercaptoethanol, and 0.5 mM EDTA) and carefully mixed through pipetting. The mixture was then centrifuged (10 min, 14,000 rpm, 4 °C), and the supernatant was collected for enzymatic activity assays. All enzyme activities were measured in the cell-free extracts of the PCP mutants after 2 h transfer of the mycelia to 25 mM D-glucose, L-arabinose, or D-xylose. Two technical replicates were performed on biological triplicates, and these were averaged in the graphs. Enzyme activities were normalized based on the total protein content of cell-free extracts, which was measured using the Pierce^TM^ BCA Protein assay kit (Thermo Fisher Scientific). Reductase activities were measured with the detection of NADPH depletion using 50 mM Na-phosphate buffer at pH 7.0, 0.2 mM NADPH, and 100 mM L-arabinose or D-xylose. We used 60 μL of cell-free extracts per reaction in a final volume of 200 μL for the reductase activity assay. Absorbance changes for NADPH at 340 nm (ε = 6.22 mM^−1^ cm^−1^) were monitored spectrophotometrically at 30 °C in clear flat-bottom 96-well plates using a FLUOstar^®^ Optima plate reader (BMG Labtech, Ortenberg, Germany).

### 2.9. Statistical Analysis

A two-tailed distribution *t*-test was conducted for the biomass and enzyme activities of cell-free extracts to compare the triple deletion mutant to the gene swap mutants grown on D-xylose and L-arabinose. Based on three biological replicates for each dataset, the t-test was evaluated with Microsoft Excel LTSC Professional plus 2021 and parameters including the two-sample equal variance. Further, the *t*-test score was expressed as a *p*-value, with the main assumption that if the value is <0.05, the datasets are significantly different (marked in Figure 3 as an asterisk *).

## 3. Results

### 3.1. Identification of Five Additional Putative Sugar Reductases

Five putative paralogs of pentose reductases LarA, XyrA, and XyrB in *A. niger* were identified using a genome-mining strategy through a BlastP search against 11 genome-sequenced fungal species using a cutoff value of E^−10^. The amino acid sequences of the resulting genes were used to assemble a phylogenetic tree to highlight the diversity and similarities within the proteins (Appendix A). As the five putative pentose reductases are positioned between the known pentose reductases of *A. niger* (LarA, XyrA, XyrB), the genes were named sugar reductases A–E (*srdA*, *srdB*, *srdC*, *srdD*, *srdE*). They were analyzed in more detail by evaluating their orthologs in other fungal species and comparing with available expression profiles in selected species.

In most clusters, conservation of expression was observed between the orthologs, such as in the clusters that contained *A. niger xyrA*, *larA*, *srdB*, *srdD*, and *srdE* (Figure 1). Exceptions to this were the *srdC* cluster, in which the *P. subrubescens* ortholog was barely expressed in any condition, and the *srdA* cluster, in which the *A. niger* ortholog was not expressed on D-galacturonic acid. Paralogs within the identified clusters showed more distinct expression profiles, as can, e.g., be seen for the *larA* and *srdE* clusters (Figure 1). Not all *A. niger srd* genes had orthologs in all tested species (Figure 1, Appendix A), indicating a high diversity in the expansion of this gene family. Orthologs in the two Sordariomycete species with expression data, *T. reesei* and *N. crassa*, were found in nearly all clusters, with an overall high expression of the genes on various sugars, except for the *srdB/srdC* cluster (Figure 1). This suggests that there is considerable variability in the evolution and expansion of this particular gene family across different species. The expression of *larA*, *xyrA*, and *xyrB* and their orthologs was the highest on L-arabinose and/or D-xylose, showing a clear connection to the PCP (Figure 1). While the expression of *A. niger* x*yrA* and *xyrB* was highly specific for D-xylose and L-arabinose, their orthologs in other fungal species also showed strong induction on other substrates, suggesting that their physiological roles may be wider than that of the *A. niger* enzymes.

The expression of the five *A. niger srd* genes was more diverse with only high expression on several substrates for *srdC* and *srdD*. While this was also the case for the *srdD* orthologs, the *srdC* ortholog was poorly expressed on all substrates. The remaining three *srd* genes (*srdA, srdB*, and *srdE*) in *A. niger* had notably lower expression. However, they did respond to specific substrates, *srdA* and *srdB* showed some expression in the presence of D-galacturonic acid, and *srdE* was less expressed in the presence of D-glucuronic acid. The expression patterns for *srdA*, *srdB*, and *srdE* in *A. niger* were not seen in their orthologs. This suggests evolutionary divergence in the function or regulation of these genes in other species compared to *A. niger*.

The differences in expression profiles within ortholog clusters suggests that the corresponding enzymes may have obtained different functions in these species, possibly adding further to the diversity of primary metabolism.

### 3.2. Biochemical Characterization of Putative Sugar Reductases

The five novel genes showed no-to-low expression on L-arabinose and D-xylose, which may explain why the corresponding enzymes did not stimulate growth on these sugars in the Δ*larA*Δ*xyrA*Δ*xyrB* strain. To determine whether they are capable of performing the conversion of L-arabinose and/or D-xylose, we determined their in vitro substrate specificity. The *A. niger* putative sugar reductases were recombinantly produced in *E. coli* and purified (Appendix A). Their activity was tested on a range of sugars and compared to LarA and XyrB (Table 2).

The new sugar reductases of *A. niger* were all able to convert D-xylose and L-arabinose (Table 2, Appendix A). XyrB was the most active reductase on D-xylose, but several of the new reductases (SrdC–SrdE) had higher activity on this substrate compared to LarA. In contrast, only SrdA had similar activity as LarA on L-arabinose, while SrdE’s activity was similar to that of XyrB. With respect to their activity on the alternative chiral form of these sugars, the activity of the eight enzymes was less conserved, with only SrdB and LarA being highly active on L-xylose, while most of them had poor activity on D-arabinose.

LarA and SrdB were not active on C6 sugars, indicating that they are pentose specific. SrdA, SrdB, and SrdD were not active on D-ribose, which suggests that they cannot accommodate a similar orientation of the OH groups at C2, C3, and C4. All the reductases that were active on C6 sugars showed activity on D-galactose, which is unique in its OH positioning of C3 and C4. Interestingly, D-lyxose, a rare pentose in nature [26], was converted by all enzymes except XyrB and was the substrate with the highest activity for SrdA and SrdC. Even though LarA is described as an L-arabinose reductase [11], it was in fact more active on L-xylose and D-lyxose, suggesting that it may have other physiological roles. Overall, the diverse specificity profiles of the reductases suggest that they also are involved in other sugar metabolic pathways. No activity was found for any of the reductases when NADPH was replaced with NADH.

Kinetic analysis of the enzymes was performed with D-xylose and L-arabinose (Table 3). Overall, the new reductases exhibited higher affinity and efficiency for L-arabinose compared to XyrB and LarA. SrdD demonstrated the lowest K_m_ and the highest k_cat_/K_m_ values, suggesting that it is the most efficient enzyme for L-arabinose. SrdE displayed the lowest K_m_ and highest catalytic efficiency for D-xylose, surpassing even XyrB in these aspects. The affinity and efficiency of the five new enzymes for D-xylose were much more diverse than those for L-arabinose. These findings underscore significant variations in the affinity and catalytic efficiency of these enzymes for pentose sugars.

### 3.3. Functional Complementation of the Triple Deletion Strain

The biochemical characterization of the novel *A. niger* sugar reductases showed that these enzymes are all able to convert D-xylose and L-arabinose. In order to evaluate whether they can functionally replace in vivo the three original reductases (XyrA, XyrB, and LarA) in vivo, they were placed between the promoter and the terminator of these original genes in strains in which the original genes were deleted. This had the added benefit that no additional copies in random genomic locations were introduced that could have indirect effects on growth (by, e.g., disrupting an important gene) or that could result in titration of the transcriptional activators controlling these genes. The resulting *A. niger* strains were compared, together with several already available deletion strains, for their growth on relevant substrates (Figure 2). Double deletion of *larA*/*xyrB* and *xyrA*/*xyrB* did not show a different phenotype compared to the reference strain, but double deletion of *larA*/*xyrA* showed reduced growth on the pentoses (Figure 2). No phenotypic difference was observed for any of them on crude substrates (Appendix A).

These double deletion strains were used to swap the remaining original reductase gene for each of the new reductases to determine whether the promoter had an influence on their in vivo role. This effectively created strains in which all three original genes were deleted but that contained one of the new *srd* genes expressed under the promoter of *larA*, *xyrA*, or *xyrB*. The Δ*xyrA*Δ*xyrB*Δ*larA* strain itself exhibited no growth on D-xylose and L-arabinose (Figure 2), so any growth observed in the generated strains must be due to the activity of the new reductase.

When any of the five new reductases was placed in either the *xyrA* or the *larA* locus, the resulting strains grew comparable to their parent strain, suggesting that each of the genes can fully compensate for the loss of *xyrA* or *larA* if expressed under that gene’s promoter. A different pattern was observed for the *xyrB* locus. While strains in which *srdA* or *srdE* was placed in the *xyrB* locus were phenotypically similar to the parental strain, strains with *srdB*, *srdC*, or *srdD* in this locus showed improved growth compared to the parental strain. In fact, the phenotype of these strains was similar to that of strains in which these genes are under control of the *xyrA* or *larA* promoter. This suggests that *srdB*, *srdC*, and *srdD* result in a more efficient PCP than *xyrB* itself. To exclude the role of the native copy of *srdB*, *srdC*, and *srdD* in this improved phenotype, the native copy was deleted in the swapped strains. However, this did not affect the phenotype (Appendix A).

Considering the surprising phenotype when some of the genes were placed in the *xyrB* locus, the effect of this on biomass production in *A. niger* liquid cultures was also evaluated (Figure 3). Surprisingly, while on plates, no significant difference was observed between the reference strain and the single mutants and all mutant strains produced much less biomass in liquid culture compared to the reference strain, suggesting that all three original genes contribute significantly to growth on D-xylose and/or L-arabinose in liquid culture.

In contrast to the plate cultures, significant differences in biomass formation were observed in the liquid cultures of the strains in which the new genes were expressed behind the promoters of *larA*, *xyrA*, or *xyrB*. However, these differences depended on the promoter behind which these genes were expressed. In the strains in which the new genes were expressed behind the *xyrA* promoter, biomass formation on D-xylose was significantly reduced for *srdA*, *srdB*, and *srdC*, while the strains with *srdD* and *srdE* had similar biomass as the parental strain (Figure 3). On L-arabinose, all strains except the one with *srdA* showed reduced biomass compared to the parental strain. When the genes were expressed behind the *larA* promoter, biomass formation on L-arabinose was highly similar, while on D-xylose, it was highly variable. Biomass formation of all the strains in which the genes were introduced behind the *xyrB* promoter (including the parental strain) was much lower on both substrates compared to the other strains. A clear difference was observed between the biomass formation in liquid culture and the growth profile on plates. While insertion of three of the genes (*srdB*, *srdC*, *srdD*) behind the *xyrB* promoter resulted in overcompensation for the loss of *xyrB* in the growth profiles on agar plates and they grew similarly as the reference strain, in liquid culture, they produced much less biomass. Also, the pattern between the different gene insertions was different, with the highest biomass formation in strains that had *srdA*, *srdC*, or *srdE* behind the *xyrB* promoter.

The only difference between the strains was the individual reductase that was present behind each promoter, and we therefore questioned whether the pentose reductase activity levels can explain the difference in biomass formation. To evaluate this, we measured both L-arabinose and D-xylose reductase activities in the mycelium of the cultures used for biomass determination of the strains in which the *srd* genes were placed behind the *xyrB* promoter (Appendix A). The activities were normalized on the total amount of intracellular proteins, as we assumed that the specific enzymes are only a small portion of this. Interestingly, while the biomass in the mutant strains was more than 2-fold reduced compared to the reference strain, the difference in the pentose reductase activity was much lower. Overall, the variation in biomass production and pentose reductase activity did not correlate for any of the strains, suggesting a more sophisticated relationship between the production of these enzymes and the biomass formation in liquid cultures.

## 4. Discussion

D-xylose and L-arabinose reductases play a crucial role in the fungal conversion of pentose sugars through the PCP. These enzymes have attracted significant interest due to their applications in the food, feed, and biofuel industries. A previous study [2] demonstrated that the initially described pathway for D-xylose and L-arabinose metabolism in *A. niger* [3] is more complex and involves multiple enzymes for most metabolic steps. The three reductases that are involved in D-xylose and L-arabinose conversion in the PCP are paralogs based on their gene-sequence- and amino-acid-sequence-based phylogeny.

Genome mining revealed five additional pentose reductase paralogs in *A. niger*, SrdA–SrdE, which were analyzed in detail in this study. A higher number of homologs of XyrA, XyrB, and LarA were revealed in Eurotiomycete fungi than in other fungi, suggesting a more expanded metabolic toolbox in Eurotiomycetes. Such a high versatile metabolic ability may be one of the reasons why these widely distributed fungi can inhabit many different biotopes and use a broad range of substrates as the carbon source, including complex biomass substrates.

All new sugar reductases identified were active on D-xylose and L-arabinose, in addition to other pentoses. Kinetic analysis of these enzymes indicated their superior performance on D-xylose and/or L-arabinose compared to XyrB and LarA. A comparison of these enzymes with another characterized pentose reductase in *Magnaporthe oryzae* PRD1, a paralog of XyrA, revealed further diversity in this group of enzymes, as, unlike the *A. niger* reductases, PRD1 has similar specific activity for L-arabinose and D-xylose [27]. This suggests that also among sugar reductases of other fungi, their specific activity toward various sugars is likely to be highly variable.

The poor expression of the *srdA–srdE* genes during *A. niger* growth on L-arabinose and D-xylose likely explains why these genes were not able to rescue the phenotype of Δ*xyrA*Δ*xyrB*Δ*larA* on these substrates. However, all these new genes could replace in vivo one or even all the original three pentose reductase genes when expressed under a different promoter, indicating that these genes are not pseudogenes. This suggests that they may play a role in the PCP in *A. niger* under conditions that induce their native expression. A thorough examination of a larger transcriptome dataset did not indicate significant expression of these genes during growth on plant-biomass-derived monosaccharides, disaccharides, polysaccharides, or crude substrates. It is possible that the inducing conditions of the new genes could, e.g., relate to specific physiological circumstances (e.g., temperature, water activity, pH) under which the original three enzymes may not be sufficiently active to secure efficient conversion of pentoses through the PCP.

Alternatively, the actual function of these genes may be related to other metabolic pathways, but this requires further study. In this context, it is worth noting that all reductases in this study belong to the aldo-keto reductase (AKR) family PF000248. Members of PF000248 exhibit remarkable catalytic versatility, playing essential roles in various cellular processes. They participate in metabolizing sugars, such as D-xylose and L-arabinose, detoxifying aldehydes, and synthesizing signaling molecules. The widespread presence of PF000248 members across diverse taxa underscores its evolutionary importance and fundamental role in biological processes [28,29,30]. An interesting finding is the high specific activity toward D-lyxose of most new *A. niger* reductases. D-lyxose is a rare pentose in nature, but so far, no pathway for this sugar involving a reductase has been reported. In bacteria, the pathway to D-lyxose involves an isomerase [31,32,33,34], but pathways in bacteria and fungi are often not conserved.

Interestingly, while the *A. niger* growth phenotypes on agar plates were highly conserved between the strains, the liquid cultures showed high variability. This suggests that the role of the sugar reductases is highly dependent on the type of cultivation and their encoding genes are possibly subject to different regulation on solid and liquid cultivation. Differences between solid and liquid cultures have been described before. A comparison of solid and liquid cultures of *A. niger* using sugar beet pulp as a substrate revealed distinct gene expression profiles for the two cultivation methods [35]. The underlying mechanism for this is currently unclear but could be related to morphology or differences in environmental conditions (e.g., aeration, oxygen diffusion rate, substrate availability). While the previous study investigated overall transcriptomic patterns on solid and liquid media, the results of our study demonstrated that the cultivation method even affects the influence of specific genes on the growth of *A. niger*.

## 5. Conclusions

The development of the biobased economy has brought about a strongly increasing interest in fungal cell factories as alternatives to conventional chemical processes. Nevertheless, fine-tuning the production of metabolic enzymes or redirecting pathways toward valuable product accumulation, even in seemingly simple pathways, like the PCP, and well-studied fungi, like *A. niger*, proves immensely challenging, and a deeper understanding of metabolism is required. Our research unveiled additional sugar reductases that expand our metabolic toolkit for synthetic biology and metabolic engineering. The diversity of these enzymes underscores the complexity of central carbon metabolism and the challenges in engineering the pathway for novel and valuable accumulation products.

## Figures and Tables

**Figure 1 jof-09-01138-f001:**
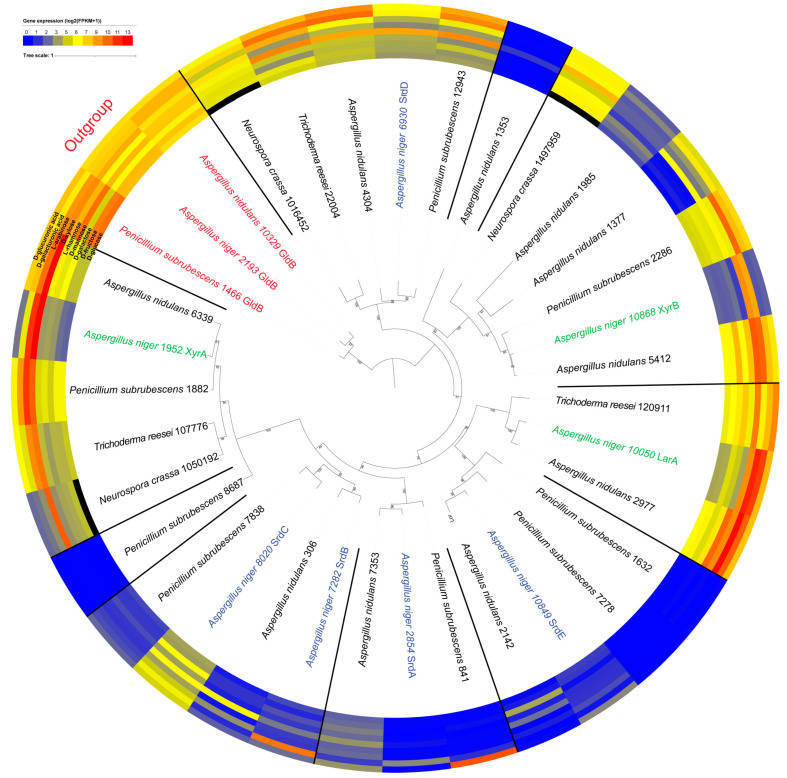
Phylogenetic analysis with heat map of available transcriptome data [19] of *A. niger* LarA, XyrA, and XyrB (green); putative pentose reductases (blue); and their homologs of selected fungal species on 25 mM D-glucuronic acid, D-galacturonic acid, L-arabinose, D-xylose, L-rhamnose, D-mannose, D-galactose, D-fructose, and D-glucose (*Neurospora crassa* transcriptome data for D-glucose are not available [20]; black field). The figure is representative of a maximum likelihood tree (500 bootstraps) of a MAFFT alignment of the amino acid sequences. The GldB outgroup is highlighted in red, and the identified clusters are separated with black lines for visualization.

**Figure 2 jof-09-01138-f002:**
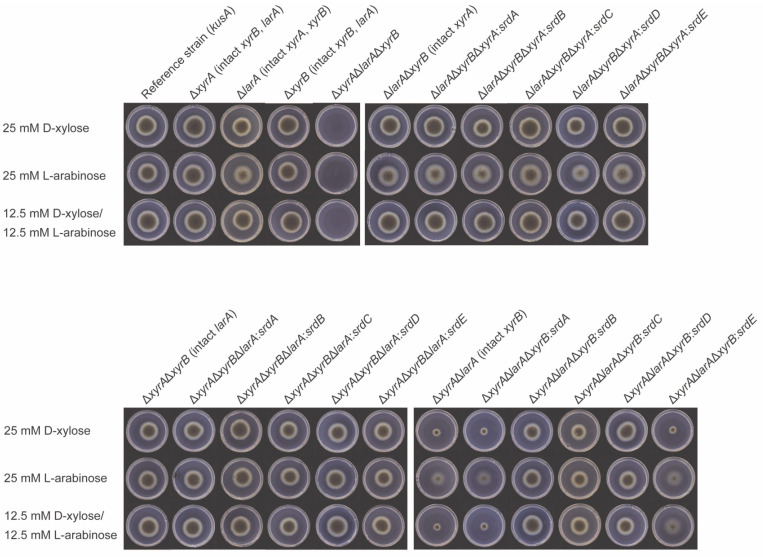
Growth profile of *A. niger* strains on D-xylose, L-arabinose, and a mixture of them after 5 days: reference (Δ*ku70*), single deletion strains (Δ*larA*, Δ*xyrB*, and Δ*xyrA*), triple deletion (Δ*larA*Δ*xyrB*Δ*xyrA*) strain, and double deletion strains (Δ*larA*Δ*xyrB*, Δ*xyrA*Δ*xyrB*, and Δ*xyrA*Δ*larA*) with swaps of the additional reductases *srdA*, *srdB*, *srdC*, *srdD*, or *srdE*.

**Figure 3 jof-09-01138-f003:**
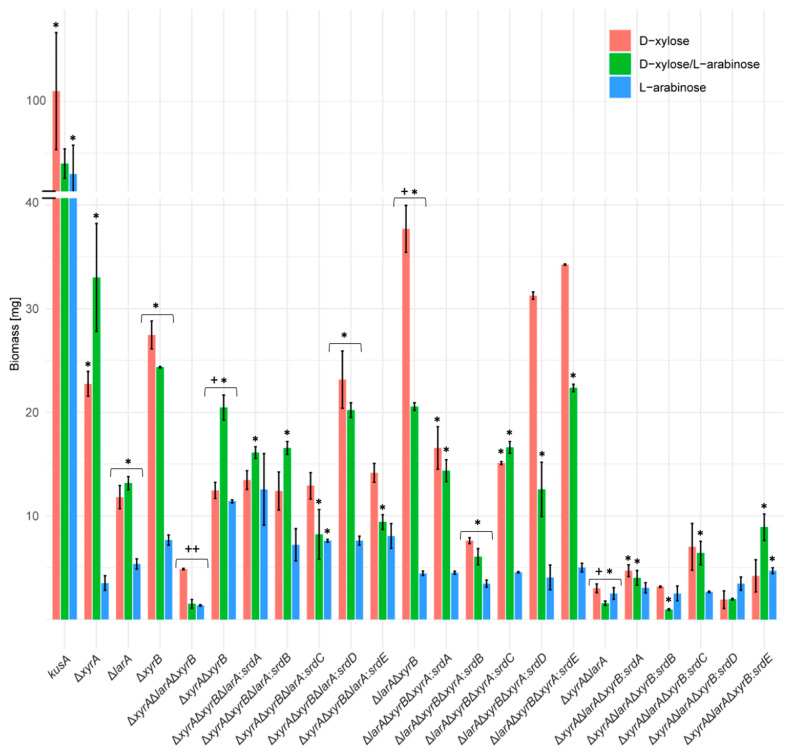
Accumulation of biomass (mg) of the *A. niger* reference strain, single deletion strains, triple deletion strain, and double deletion strains with putative gene swaps in liquid cultures on 25 mM D-xylose, L-arabinose, and D-xylose/L-arabinose after 4 days of cultivation. ++, statistical reference for single and double deletion strains; +, statistical reference for putative gene swaps into the corresponding double deletion strain; *, *p* < 0.05.

**Table 1 jof-09-01138-t001:** *A. niger* strains used in this study.

CBS Number	Genotype	Reference
CBS 138852	*cspA1*, *kusA::amdS*, *pyrA6* (reference strain)	[21]
CBS 144530	*cspA1*, *kusA::amdS*, *pyrA6*, Δ*xyrA*, Δ*xyrB*, Δ*larA*	[2]
CBS 144040	*cspA1*, *kusA::amdS*, *pyrA6*, Δ*larA*	[2]
CBS 144535	*cspA1*, *kusA::amdS*, *pyrA6*, Δ*xyrB*	[2]
CBS 144037	*cspA1*, *kusA::amdS*, *pyrA6*, Δ*xyrA*,	[2]
CBS 144043	*cspA1*, *kusA::amdS*, *pyrA6*, Δ*xyrB*, Δ*larA*	[2]
CBS146916	*cspA1*, *kusA::amdS*, *pyrA6*, Δ*xyrA*, Δ*xyrB*	[2]
CBS 150433	*cspA1*, *kusA::amdS*, *pyrA6*, Δ*xyrA*, Δ*larA*	This study
CBS 150434	*cspA1*, *kusA::amdS*, *pyrA6*, Δ*xyrA*:*srdA*	This study
CBS 150435	*cspA1*, *kusA::amdS*, *pyrA6*, Δ*xyrA*:*srdB*	This study
CBS 150436	*cspA1*, *kusA::amdS*, *pyrA6*, Δ*xyrA*:*srdC*	This study
CBS 150437	*cspA1*, *kusA::amdS*, *pyrA6*, Δ*xyrA*:*srdD*	This study
CBS 150438	*cspA1*, *kusA::amdS*, *pyrA6*, Δ*xyrA*:*srdE*	This study
CBS 150439	*cspA1*, *kusA::amdS*, *pyrA6*, Δ*xyrB*:*srdA*	This study
CBS 150440	*cspA1*, *kusA::amdS*, *pyrA6*, Δ*xyrB*:*srdB*	This study
CBS 150441	*cspA1*, *kusA::amdS*, *pyrA6*, Δ*xyrB*:*srdC*	This study
CBS 150442	*cspA1*, *kusA::amdS*, *pyrA6*, Δ*xyrB*:*srdD*	This study
CBS 150443	*cspA1*, *kusA::amdS*, *pyrA6*, Δ*xyrB*:*srdE*	This study
CBS 150444	*cspA1*, *kusA::amdS*, *pyrA6*, Δ*larA*:*srdA*	This study
CBS 150445	*cspA1*, *kusA::amdS*, *pyrA6*, Δ*larA*:*srdB*	This study
CBS 150446	*cspA1*, *kusA::amdS*, *pyrA6*, Δ*larA*:*srdC*	This study
CBS 150447	*cspA1*, *kusA::amdS*, *pyrA6*, Δ*larA*:*srdD*	This study
CBS 150448	*cspA1*, *kusA::amdS*, *pyrA6*, Δ*larA*:*srdE*	This study

**Table 2 jof-09-01138-t002:** Summary of sugar specificity of the new *A. niger* pentose reductases SrdA–SrdE and previously characterized pentose reductases XyrB and LarA. The values represent the average activity of triplicate assays, and the standard deviation is provided alongside each value. The activity assays were performed with 100 mM substrate, 2 mM NADPH at pH 7.

Substrate	SrdA (U mg^−1^)	SrdB (U mg^−1^)	SrdC (U mg^−1^)	SrdD (U mg^−1^)	SrdE (U mg^−1^)	LarA (U mg^−1^)	XyrB (U mg^−1^)
D-xylose	12.4 ± 0.4	7.5 ± 1.4	31.3 ± 17.7	37.1 ± 5.5	38.5 ± 1.4	25.0 ± 2.2	103.5 ± 3.4
L-arabinose	39.3 ± 2.7	8.8 ± 1.3	25.9 ± 2.8	20.2 ± 3.2	88.2 ± 1.1	40.9 ± 8.6	87.7 ± 3.1
L-xylose	21.9 ± 3.2	30.3 ± 8.9	24.0 ± 2.5	15.2 ± 0.4	n.d.	41.2 ± 7.2	25.2 ± 2.9
D-arabinose	12.9 ± 1.3	25.7 ± 3.9	22.3 ± 2.5	n.d.	n.d.	10.6 ± 1.3	n.d.
D-ribose	n.d.	n.d.	21.9 ± 8.5	n.d.	42.9 ± 3.9	23.8 ± 4.0	33.1 ± 1.6
D-lyxose	118.9 ± 7.8	12.6 ± 0.5	40.9 ± 2.6	14.8 ± 4.2	59.1 ± 8.6	48.6 ± 3.6	n.d.
D-glucose	n.d.	n.d.	17.4 ± 1.2	n.d.	21.2 ± 7.0	3.7 ± 2.6	20.3 ± 0.5
D-fructose	n.d.	n.d.	n.d.	n.d.	25.6 ± 2.0	n.d.	n.d.
L-rhamnose	n.d.	n.d.	n.d.	14.4 ± 3.6	56.6 ± 25.8	n.d.	n.d.
D-galactose	17.2 ± 2.0	n.d.	16.5 ± 11.4	15.2 ± 0.1	18.3 ± 5.9	n.d.	88.9 ± 3.9
L-sorbose	n.d.	n.d.	25.2 ± 10.4	19.2 ± 4.0	20.7 ± 1.3	n.d.	21.2 ± 0.5
D-mannose	16.7 ± 0.5	n.d.	n.d.	12.6 ± 4.8	12.9 ± 2.0	n.d.	22.2 ± 7.2

n.d.: not detected.

**Table 3 jof-09-01138-t003:** Comparison of kinetic constants of the five new sugar reductases SrdA–SrdE, LarA, and XyrB of *A. niger* on D-xylose and L-arabinose.

	D-xylose			L-arabinose		
Protein	K_m_ (mM)	k_cat_ (s^−1^)	k_cat_/K_m_ (s^−1^mM^−1^)	K_m_ (mM)	k_cat_ (s^−1^)	k_cat_/K_m_ (s^−1^mM^−1^)
**SrdA**	88.5	31.7	0.4	4.7	27.7	5.9
**SrdB**	2.8	22.3	8.0	4.5	27.3	6.1
**SrdC**	33.3	3.1	0.1	2.3	21.5	9.4
**SrdD**	3.4	19.5	5.7	0.9	17.7	19.6
**SrdE**	1.1	19.0	17.3	4.3	25.1	5.8
**XyrB**	3.0	11.2	3.7	12.3	8.4	0.7
**LarA**	10.3	1.8	0.2	12.5	4.6	0.4

## Data Availability

The reads from each of the transcriptome sequencing (RNA-seq) were previously published [19] and samples were deposited in the Sequence Read Archive at NCBI under the accession numbers: *A. niger* SRP448993, SRP449003–SRP449007, SRP449023, SRP449039, SRP449049, SRP449062, SRP449079–SRP449081, SRP449083–SRP449085, SRP449089, SRP449068–SRP449070, SRP449098, SRP449125, SRP449138, SRP449141, SRP449142, SRP449151, and SRP449193; *A. nidulans* SRP262827-SRP262853; *P. subrubescens* SRP246823-SRP246849; and *T. reesei* SRP378720-SRP378745. N. crassa transcriptome data was used from Wu et al., 2020 [20] and have been deposited National Center for Biotechnology Information (NCBI) Sequence Read Archive (ID SRP133337).

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
