# Peer review of "Genome Mining Reveals a Surprising Number of Sugar Reductases in *Aspergillus niger"

_jof, 2023, doi:10.3390/jof9121138_

Round 1

Reviewer 1 Report

Comments and Suggestions for Authors

Müller et al., investigated the activity and functional complementation of five potential paralogs of three pentose reductases in Aspergillus niger. However, this current manuscript could not be accepted for publication before address the issues as follows.

1.      Is it necessary to identify more pentose reductases in addition of three major pentose reductases? The triple deletion mutant cannot grow on the media using D-xylose and L-arabinose as solo carbon source. Please provide more background and discussion to highlight the purpose and its research significance of this manuscript.

2.      Line 314-317: “strains with srdB, srdC or srdD in this locus show improved growth compared to the parental strain”, “srdB, srdC and srdD result in a more efficient PCP than xyrB itself”. This finding is too superficial. Please confirm the function of srdB, srdC and srdD by constructing their triple deletion mutants using ΔlarAΔxyrA as host. In addition, please supplement the intracellular PCP intermediate metabolites, PCP enzyme activities and expression profiles of genes involved in pentose catabolism of these mutants including these triple deletion mutants.

Comments on the Quality of English Language

Please check the language of this manuscript throughout.

Author Response

  1. Is it necessary to identify more pentose reductases in addition of three major pentose reductases? The triple deletion mutant cannot grow on the media using D-xylose and L-arabinose as solo carbon source. Please provide more background and discussion to highlight the purpose and its research significance of this manuscript.

> Response: The initial reason for this was the minimal phenotype on crude pentose containing substrates. The analysis of the A. niger genome then revealed additional paralogs of the initial three genes that were not or poorly expressed on L-arabinose and D-xylose, which made us consider if they could be backup genes under more complex conditions. We therefore studied their functionality in vitro and in vivo. The text has been modified to make this more clear.

  1. Line 314-317: “strains with srdBsrdCor srdD in this locus show improved growth compared to the parental strain”, “srdB, srdC and srdD result in a more efficient PCP than xyrB itself”. This finding is too superficial. Please confirm the function of srdB, srdC and srdD by constructing their triple deletion mutants using ΔlarAΔxyrA as host. In addition, please supplement the intracellular PCP intermediate metabolites, PCP enzyme activities and expression profiles of genes involved in pentose catabolism of these mutants including these triple deletion mutants.

       > Response: We don’t really understand this comment for a number of reasons:

       - The expression of these genes on L-arabinose and D-xylose is very low (as indicated in the manuscript) so deletion of them is unlikely to have a major effect.

       - we deleted the native gene in the background in which it was inserted behind the xyrB promoter and this deletion had no effect, demonstrating that the phenotype is due to the insertion, This also demonstrates that the other two genes in their native locus do not effect this.

       - Pentose reductase activity is already measured for these strains and included in the manuscript.

            We don’t think that measuring the expression profiles of the other genes will provide additional insights as the promoter sequence has not been altered and no additional copies have been inserted, so it is highly unlikely that there would be any titration of the regulator(s) activating the expression of these genes. We chose the current approach to specifically avoid that. In light of this, I think our suggestion is warranted and the most likely explanation of the phenotype observed.

In addition, english was checked by a native speaker and modified where needed.

Reviewer 2 Report

Comments and Suggestions for Authors
The authors of the Manuscript “Genome mining reveals a surprising number of sugar reductases in Aspergillus niger” presents a consistent line of research that led to the identification of several new sugar reductases.
However, several points should also be revised in the manuscript before publication.
 Major:
Data transparency. The expression data related to this study should be either be made publicly available if not done so already or the proper references should be given to allow the reader to source the transcriptomic data related to the study. There is not a single line of text describing the processing workflow of the presented transcriptome data (figure 1) or their source in this manuscript. The authors are highly encouraged to change this.

Differences of growth between solid culture and liquid culture: Why did the authors did not perform enzyme activity tests for strains where the xyrA promoter was used to drive gene expression of sugar reductase genes? Choosing the direct enzyme activity assays is perhaps much more relevant here than looking into expression data of the constructs, but what is known about the expression of all genes of this study in liquid culture? Would this correlate to the observed enzyme activities of the biomass?
 Minor
Figure 1: Give the reference of the used transcriptomics data
 2.8. Preparation of cell free extracts and enzyme activity assays – Briefly describe how cells where disrupted
Figure 3 – finalize the concentration of the used substrates (replace the XX) . According to the method section, this are 25 mM of each substrate?
Line 334 onwards  - description of results presented in figure 3: Why did the authors choose to consider the biomass increase resulting from expression of srdD under control the xyrA promoter significant whilst the biomass formation on D-xylose is even higher for srdE?
 Line 346 onwards: Please rewrite that sentence to make clear to the reader that there is a discrepancy between the genes observed as compensating the growth defect between liquid culture and agar plates. The sentence  is very confusing for the reader.

Author Response

Major:

Data transparency. The expression data related to this study should be either be made publicly available if not done so already or the proper references should be given to allow the reader to source the transcriptomic data related to the study. There is not a single line of text describing the processing workflow of the presented transcriptome data (figure 1) or their source in this manuscript. The authors are highly encouraged to change this.

> Response: the data used in the study has been previously published previously. We have added the citations in the materials and methods and in Figure 1.

Differences of growth between solid culture and liquid culture: Why did the authors did not perform enzyme activity tests for strains where the xyrA promoter was used to drive gene expression of sugar reductase genes? Choosing the direct enzyme activity assays is perhaps much more relevant here than looking into expression data of the constructs, but what is known about the expression of all genes of this study in liquid culture? Would this correlate to the observed enzyme activities of the biomass?

> Response: The expression data referred to above is from liquid culture, so this does directly relate to the culture conditions in the current study. We have in fact performed the pentose reductase assays for these strains as included in the manuscript. Considering that the main pentose reductases have been removed in these strains, the activity measured in the mycelium can be directly related to the inserted gene. We have modified the text to clarify this better.

Minor

Figure 1: Give the reference of the used transcriptomics data

> This has been added

 2.8. Preparation of cell free extracts and enzyme activity assays – Briefly describe how cells where disrupted

> This has been added

Figure 3 – finalize the concentration of the used substrates (replace the XX) . According to the method section, this are 25 mM of each substrate?

> This has been added

Line 334 onwards  - description of results presented in figure 3: Why did the authors choose to consider the biomass increase resulting from expression of srdD under control the xyrA promoter significant whilst the biomass formation on D-xylose is even higher for srdE?

> Response: We appreciate the reviewer for pointing this out and have modified the text to better reflect the data.

 Line 346 onwards: Please rewrite that sentence to make clear to the reader that there is a discrepancy between the genes observed as compensating the growth defect between liquid culture and agar plates. The sentence is very confusing for the reader.

> This has been rewritten

Round 2

Reviewer 1 Report

Comments and Suggestions for Authors

The current manuscript could be accepted for publication in JoF.